# Student Experience and Satisfaction in Academic Libraries: A Comparative Study among Three Universities in Wuhan

Lei Peng [1,2] 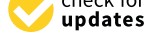, Wenli Wei [1,2], Wenyi Fan [1,2,*], Shitao Jin [1,2] and Yuxuan Liu [1,2]

1 School of Architecture & Urban Planning, Huazhong University of Science and Technology, Wuhan 430074, China; penglei@hust.edu.cn (L.P.); m201973504@hust.edu.cn (W.W.); m202073616@hust.edu.cn (S.J.); m202073609@hust.edu.cn (Y.L.)
2 Hubei Engineering and Technology Research Center of Urbanization, Wuhan 430074, China
* Correspondence: fan_wenyi@hust.edu.cn

**Abstract:** In recent years, the spatial renovation of university libraries in various countries has focused on readers' needs and followed the trend to develop learning spaces as a primary spatial form. In this study, we reviewed six spatial dimensions affecting student users' learning experience. Specifically, we built a theory- and practice-based conceptual analysis framework to measure users' satisfaction with recent spatial renovations at three university libraries in Wuhan, China. We used SPSS statistical software to conduct multiple linear regression analyses of spatial satisfaction. The findings show that five spatial dimensions significantly affect students' satisfaction with library space, namely, service facility availability, quality of interior design, physical environment elements, spatial diversity, and learning space controllability. Service facility availability is the most critical factor affecting spatial satisfaction. In this study, we present empirical, evidence-based space elements that enhance user satisfaction with library spaces, and provide targeted design suggestions for future library space renovation and the optimization of space allocation and expansion of space services at university libraries in China.

**Keywords:** academic library; student experience; learning space; satisfaction assessment; comparison study

## 1. Introduction

Academic libraries have always been influenced by challenges and changes in higher education. The advent of the knowledge economy, the rapid development of information technology, and the evolution of learning theory have influenced how libraries develop and construct new spaces to meet the academic needs of students and faculty.

Although traditional library services, such as collections, information storage, access, and distribution systems, are necessary, information technology has revolutionized patrons' information search habits, and convenient online services have replaced retrieval and circulation services. Along with the development of constructivist learning theory on pedagogy, today's higher education emphasizes group learning and social learning. In addition, the goals of higher education have evolved from offering purely professional education to encouraging innovation and talent. Thus, the library's previous function of simple storage and one-way transmission of knowledge is no longer suitable for the mission of contemporary universities. These changes have led to the spatial renovation of libraries since the 1990s, brought the value of space to the forefront, and led to attempts to build information sharing spaces and learning sharing spaces. At the beginning of the 21st century, librarian Bennett pointed out that "libraries should be places that promote learning." After reviewing library space renovation practices in the 1990s, he said, "Teaching and learning are essential to the mission of academic libraries, and past practices have ignored the learning needs of patrons" [1]. Black and Roberts (2006) state that "by placing learners and learning at the center of service and space renewal, library and information

services can achieve a vision of a future that is vibrant, creative and based on genuine learning communities" [2]. The university library has evolved into a significant informal learning space, a true center of learning that provides a supportive environment, stimulates collaborative social learning, and encourages creativity and knowledge creation.

As philosopher Gaston Bachelard mentions in *The Poetics of Space*, space is directly related to human existence; space is not just a container for objects in the material sense, but also a happy dwelling place for human consciousness [3]. The architectural theorist Bruno Zevi also believes that the focus of architecture is the interior space, rather than partial walls or roofs. It should not be viewed in the same way as paintings or sculptures, but rather based on how people feel in the space and how they relate to each other [4]. Therefore, it is important to explore the real feelings and thoughts of library users. Research on assessment of library space renovation has been substantial in the last two decades, bringing together many POE (postoccupancy evaluation) assessments and studies involving anthropological approaches that have become a core component of the library space renovation process. Assessments focus on user experience in terms of types of learning activities, learning preferences, space-type needs, and social needs. Northeastern University carried out the earliest spatial value assessment in the US. Among the more representative assessments is the TEALS (Tool for Evaluation of Academic Library Spaces) project of Deakin University in Australia, which proposed seven recommendations for spatial renovation. The assessment project of the University of North Carolina in the US offered a progressive puzzle approach to assessing while renovating. Sheffield Hallam University in the UK proposed 10 elements of spatial renovation based on user surveys [5].

These inventory-based approaches to space assessment have provided directional guidance for space renovation and new construction at colleges and universities. After more than 20 years of development, the layout of university library space has changed significantly. However, the focus of many of these assessments varies, with some focusing on the space elements that attract students to the library and others concentrating on space utilization that does not fully correspond with the student's learning experience. From an architectural design perspective, there is still a need to explore which specific spatial design elements influence the student learning experience.

In this study, we collected feedback on and evaluation of library space performance from the perspective of students' learning experience. We gathered these data using the empirical method of a questionnaire survey to interview end users at three university libraries in Wuhan. An attempt was made to explore how architectural design can enhance and facilitate users' learning experience in libraries and identify which elements of the space impact the learning experience and which spatial factors affect users' satisfaction with the library space. Evidence-based strategic guidelines were identified for the future design of re-engineered university library spaces.

## 2. Literature Review and Research Framework

The literature review builds on the growing number of studies about the relationship between the physical learning environment and students' learning experience in university libraries and informal learning spaces. To explore how learning takes place, researchers have explored different spatial design features, and many have proposed design principles and a set of key features that contemporary library learning spaces should exhibit. In the next section, we summarized six key physical space elements gleaned from previous research that have had an impact on students' learning experience and, based on these studies, developed an analytical framework for subsequent empirical research (see Figure 1).

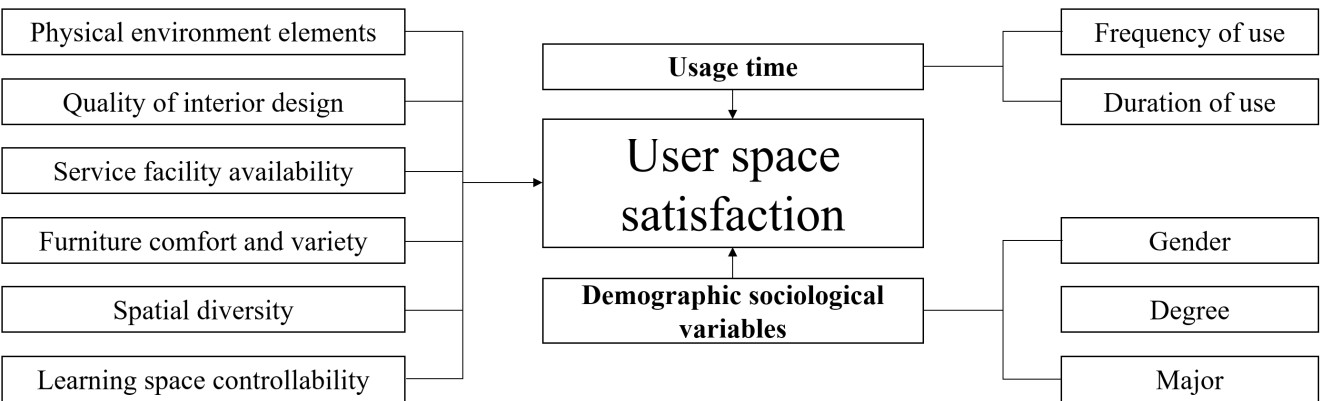

**Figure 1.** Research framework.

Many studies have found that the physical elements of the learning environment affect the learning experience and learning effectiveness. Good ventilation and a comfortable temperature deter indoor pollutants, particularly carbon dioxide. Poor indoor ventilation concentrates indoor pollutants and can lead to inefficient learning activities [1,6] and harm visual acuity, memory breadth, reaction speed, concentration, and the stability of students' movements [7]. Some evidence suggests that lighting levels affect students' moods and attitudes. Although natural light may cause an undesirable glare and solar overheating effects, in general, natural light positively impacts students by improving perception, behavior, and attention [8]. Numerous interviews and questionnaires have also demonstrated students' preference for natural light and their desire for more natural light in their learning spaces [5,9,10]. Noise level is also a critical factor for students when choosing a study space [11]. In Wu et al.'s (2021) experiment, the appropriate noise level was the most significant driver of student learning. Some students do not accept a learning environment with background noise, believing that the surrounding noise would distract them when they needed to complete stressful tasks. They understand that although small noises, such as the sounds of eating, typing, paper rustling, and coughing, are annoying and a disturbance, absolute silence is not possible in a general study area—so they bring their earplugs [5,12–14].

The results of the previous studies confirm the correlation between the quality of interior design of higher education buildings and students' learning experience. Although interior design is not the most important influence on students' choice of learning space, poorly decorated interiors can deter students from choosing to study in such spaces [11,15]. In Velusamy's (2021) study, students expressed their desire for learning spaces to have an attractive color scheme and decorative features and be connected to nature [16].

The information age has profoundly impacted students' learning behaviors. Qian (2020) found that over 60% of patrons bring their electronic devices to libraries to use for study. Thus, the availability of relevant ICT facilities and Wi-Fi coverage and power outlets have become essential infrastructure [17]. Yu (2018) found that respondents first look for the availability of outlets next to their seats when choosing a study space. It is evident that the configuration of power outlets directly affects users' learning experience and frequency of use [18]. The COVID-19 pandemic has accelerated the popularity of online courses, and online devices have become essential for students. In addition to personal computers, projection devices and digital graphic organizers (visual thinking tools that make pictures of your thoughts) enhance students' learning experience when interacting with peers [5]. Andrews (2016) found that students typically want more outlets and whiteboards than other more sophisticated technologies related to the rise of personal devices and cloud computing [9].

In addition to technical equipment, Harrop (2013) found that long open hours—24 h a day, Sunday through Thursday, were essential for some users who often spent days and nights at the learning centers. The learning center was also used more frequently in the

evening than other campus spaces [5]. Connors (2012) found that learning spaces that offered food and beverage were more attractive to learners—63% of learners at Sewanee–the University of the South in the US felt that food and beverage helped them to stay focused while studying [19]. During the user participatory design process of the McKeldin Library at the University of Maryland, students wanted lounge facilities, such as a coffee bar in the library, where they could take breaks and eat [10]. Deng (2017) mentioned that a café in the library has the potential to increase social learning in the library as it provides a comfortable place for students to meet and talk [20]. Souter (2011) found that extended visiting hours, easy access to food, lockers, and reconfigurable spaces are essential elements for enhancing the learning experience [21].

High-quality furniture facilities have been shown to impact the development of learning activities [10,22] significantly. Furniture is a key infrastructure of the learning environment and involves comfort, ergonomics, and functionality. Adjustable and movable tables and chairs can support the reconfiguration of spatial layouts anytime and anywhere. Studies by Wu (2021), Becker (2015), Harrop (2013), and others reported that adaptable and flexible furniture plays an essential role in learning spaces. For example, modular furniture that can be freely combined can support students' freedom to reconfigure spatial layouts according to their needs and preferences, thus increasing the level of student collaboration and learning [5,12,15]. A study by Andrews (2016) also highlights the vital roles of furniture and spatial layout for creating a seamless learning environment [9].

To meet the growing demand for spaces that are conducive to learning, libraries create various types of features for different purposes. In addition to traditional public reading rooms, libraries have added group study areas and small-group discussion rooms [23,24]; computer workstations [25,26]; nontraditional facilities, such as cafes and relaxed public study areas [27]; and quiet, exclusive study rooms for individual study. A current trend in the changing pedagogy of higher education is the emphasis on problem-based project learning (PPL) and thus the need for group workspaces. Separate discussion rooms are prevalent in 21st-century learning spaces because the sounds of discussion can be disruptive to other learners. Another notable change in libraries is the introduction of coffee and recreational areas for learners to rest and relax (e.g., study cafes, bookstores). Comfortable sofas and readily available refreshments provide an inviting space that encourages students to linger, meet, and talk outside of class and allows for a variety of active social learning activities [28]. The casual atmosphere of the café helps learners to refresh themselves after hard academic work. To summarize, academic libraries are primarily used for learning purposes by learners who are seeking a quiet environment in which to engage with their academic work. Many libraries have begun to add individual study rooms and individual study areas with clear spatial separation and different levels of quietness to meet learners' spatial preferences.

Contemporary college students have a greater sense of autonomy but also want a supportive public learning space. According to a behavior log study by Beckers (2016), most learning activities occur at home. Learners cited autonomy as one of the main reasons for choosing to study at home, where they could control the temperature of the room, play their favorite music, or eat while studying. This autonomy makes home the preferred learning space [15]. Compared with the austere spaces of the past, library study spaces today are now more intimate and relaxed; libraries want to create spaces that are homelike, where patrons can have some control over the space. Some students need a tranquil study atmosphere, whereas others say they prefer to study in noisy environments. These students say they like the feeling of being slightly disturbed and able to make a small amount of noise themselves, such as talking, eating, and socializing. Being in a space where they can make noise means that their learning activities are less restricted and they are free to be themselves [12,13,29]. Overall, the research findings support empowering students to have control over their learning area.

Based on the findings of the above research studies, we identified six physical spatial dimensions as follows: service facility availability, quality of interior design, physical envi-

ronment elements, spatial diversity, learning space controllability, and furniture comfort and variety. The six spatial dimensions make up the proposed analytical framework, which is shown in Figure 1. We hypothesize that all six spatial dimensions influence students' learning experiences in both individual and cooperative learning. The literature suggests that sociodemographic characteristics, such as gender, age, years of study, and frequency of visiting the library, may also influence the learning experience.

## 3. Materials and Methods

### 3.1. Case Selection

The participating universities for this study were Wuhan University (WHU), Huazhong University of Science and Technology (HUST), and Wuhan University of Technology Nanhu Campus (WUT). The reasons these universities were selected are as follows:

- Accessibility. All three universities are located in Wuhan, and the research team is from HUST in Wuhan.
- Completion of renovation after 2010. The most recent spatial redevelopment for the library at WHU was completed in 2011. The expansion of the library at HUST was completed in 2015. The building areas are 35,548 square meters and 43,959 square meters, respectively. The library of WUT was completed in 2016 and officially opened in 2018 with a floor area of 48,800 square meters.
- Representatives of space renovation. As one of the first university libraries in China to build a shared learning space, WHU has been a trendsetter in space renovation. However, it has not made any directional changes to the overall space renovation. The space renovation of the library at HUST consists of simple, functional upgrades —larger reading and leisure study areas. However, it has not considered the overall space from the perspective of students' learning experience. The library of WUT, as a newly built library, represents the latest design trend in domestic university libraries.

The three libraries are typical representatives of space renovation in China's university libraries. Our goal is to analyze the differences in students' learning experiences and space satisfactions through this comparative study (as illustrated in Tables 1 and 2).

**Table 1.** Current status of learning spaces in three universities.

| | HUST | WHU | WUT |
|---|---|---|---|
| Photo | 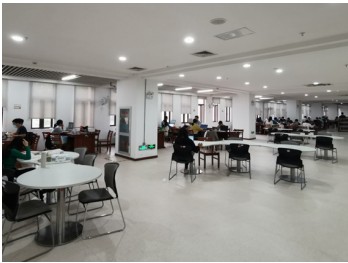 | 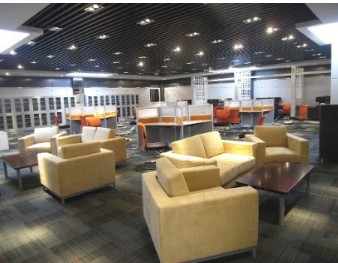 | 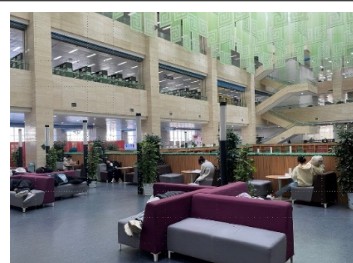 |

**Table 1.** *Cont.*

| | HUST | WHU | WUT |
|---|---|---|---|
| Floor plans | | | |
| Model | | | |

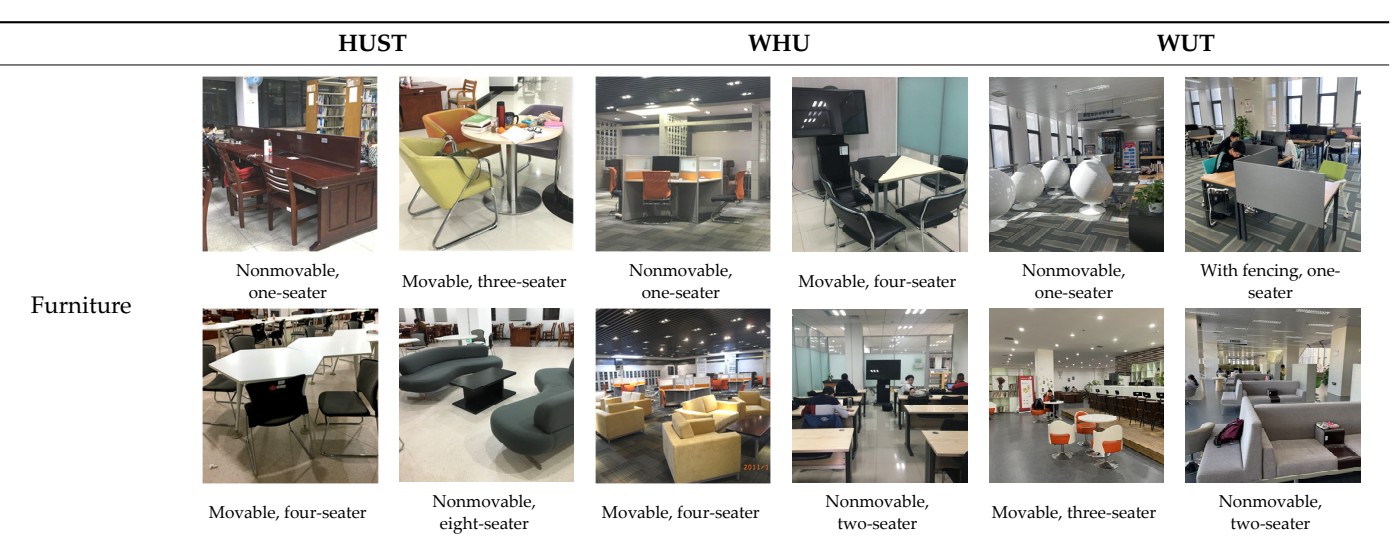

**Table 2.** Furniture and space types of learning spaces in three universities.

| | HUST | | WHU | | WUT | |
|---|---|---|---|---|---|---|
| Furniture | Nonmovable, one-seater | Movable, three-seater | Nonmovable, one-seater | Movable, four-seater | Nonmovable, one-seater | With fencing, one-seater |
| | Movable, four-seater | Nonmovable, eight-seater | Movable, four-seater | Nonmovable, two-seater | Movable, three-seater | Nonmovable, two-seater |

**Table 2.** *Cont.*

| | HUST | | WHU | | WUT | |
|---|---|---|---|---|---|---|
| Space types | <br>Personal learning area | <br>Group learning area | <br>Personal learning area | <br>Personal learning room | <br>Personal learning area | <br>Personal learning room |
| | <br>Leisure discussion area | <br>Computer learning room | <br>Group learning room | <br>Leisure discussion area | <br>Group learning area | <br>Leisure discussion area |

### 3.2. Measurement Tool

We adopted a comparative study design using quantitative methods. We used a questionnaire as the sole data collection method to reach as many respondents as possible.

The questionnaire was created in the form of a web-based completion of Sojump (a professional online questionnaire platform in China) and was distributed randomly through social networks (WeChat group and QQ group) and via face-to-face contact in the libraries from November 2021 to January 2022.

### 3.3. Survey Design

In the first part of the survey, users were asked to provide personal information, such as their gender, college year, major, and information about the frequency and length of their visits to the library. The second part of the survey questionnaire numerically ranked students' satisfaction with and attitudes to their libraries' spatial design characteristics. The answers were rated on a 5-point Likert scale (e.g., 1 = strongly dissatisfied; 2 = dissatisfied; 3 = no comment; 4 = satisfied; and 5 = strongly satisfied). The survey questions were based on the 6 dimensions and 27 design elements (see Appendix A: Questionnaire Survey).

## 4. Results

### 4.1. Respondent Characteristics

In this paper, a commonly used statistical software program (SPSS) was used as the primary tool for analyzing the questionnaires. A total of 497 questionnaires were distributed, and 486 questionnaires were recovered. After eliminating incomplete, duplicate, or invalid responses, 457 valid responses were analyzed—152 from HUST, 154 from WHU, and 151 from WUT. The scale's reliability was 0.88, and the KMO (Kaiser–Meyer–Olkin Measure, a statistic that indicates the proportion of variance in your variables that might be caused by underlying factors) value was 0.877 (significance higher than 0.8), indicating a reasonable degree of achievement. The *p*-value was $p = 0.000 < 0.05$ in Bartlett's spherical test.

The gender ratios of the users in all three schools were relatively even with male-to-female ratios of 0.924, 1.081, and 1.097, respectively. In terms of educational level distribution, the library of HUST had the highest percentage of undergraduates (50%), followed by master's degree students (47.4%), doctoral degree students (2%), and faculty members (0.7%). The distribution of the educational background of library users in WHU and WUT was consistent, with the highest percentage being master's degree students, followed by undergraduate degree students, doctoral degree students, and faculty members. Regarding the distribution of the professional disciplines of the research sample subjects,

the data results of all three universities show that the highest percentages of professional disciplines were distributed across science, engineering, agriculture, and medicine (as illustrated in Table 3).

**Table 3.** Statistical table of the sample distribution of the space satisfaction questionnaire of the libraries of the three universities.

| Title | HUST Library Total *n* = 152 | WHU Library Total *n* = 154 | WUT Library Total *n* = 151 |
|---|---|---|---|
| Gender: | | | |
| Male | 73 (48%) | 80 (51.9%) | 79 (52.3%) |
| Female | 79 (52%) | 74 (48.1%) | 72 (47.7%) |
| Degree: | | | |
| Undergraduate | 76 (50%) | 55 (35.7%) | 61 (40.4%) |
| Master's | 72 (47.4%) | 86 (55.8%) | 71 (47%) |
| Doctorate | 3 (2%) | 11 (7.1%) | 14 (9.3%) |
| Faculty and researchers | 1 (0.7%) | 2 (1.3%) | 5 (3.3%) |
| Other | 0 (0%) | 0 (0%) | 0 (%) |
| Major: | | | |
| Philosophy, economics, law | 26 (17.1%) | 15 (9.7%) | 18 (11.9%) |
| Pedagogy, literature, history | 22 (14.5%) | 24 (15.6) | 24 (15.9%) |
| Science, engineering, agriculture, medicine | 78 (51.3%) | 103 (66.9%) | 90 (59.6%) |
| Military Science, management science, science of art | 26 (17.1%) | 12 (7.8%) | 19 (12.6%) |

The statistics for all three schools show that the highest percentages of users visit the library every week—46.1% of users to the HUST library, 43.5% of users to the WUT library, and 43% of users to the Wuhan Polytechnic University library. Further cross-analysis of user frequency data and the academic level of users found that undergraduate students in the three schools visited the library more frequently than master's degree students, doctoral degree students, and faculty members.

In all three schools, the highest percentage of users are those who use the library for more than 3 h per visit. Among them, 55.3% of users of the HUST library, 47.4% of users of the WHU library, and 43% of users of the WUT library choose to use the library for more than 3 h per visit.

According to the analysis, the percentages of library users for self-study activities from all three universities are as follows: 88.2% from the HUST library, 82.5% from the WHU library, and 80.1% from the WUT library (see Figure 2).

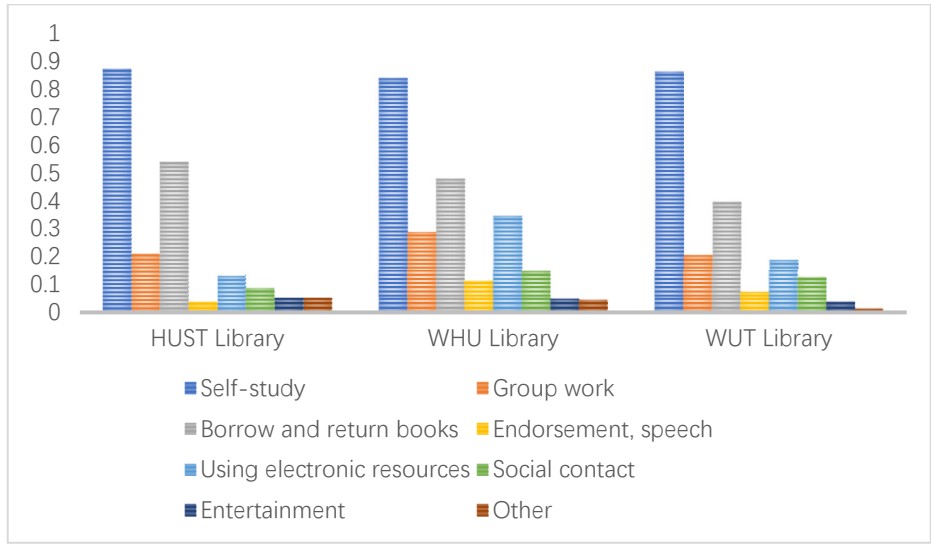

**Figure 2.** Statistical table of sample distribution.

### 4.2. Description of Satisfaction Status

Table 4 shows significant differences between the individual satisfaction scores of users at the three libraries. Users at WUT had the highest overall space satisfaction scores, indicating that this recently constructed library met students' learning needs. Quality of interior design, spatial diversity, and service facility availability received the highest scores. Physical environment elements and learning space controllability received low scores. This is due to the inadequate operation mode of the library of WUT after opening. For example, students are unable to adjust the equipment, such as air conditioning and lighting, resulting in a low rating for the environmental aspects of the physical space. However, the excellent quality of interior design and the availability of space allow students to still have good overall impressions of the spaces.

**Table 4.** Situation analysis of space satisfaction questionnaire for the three university libraries.

|  | HUST Library Mean | WHU Library Mean | WUT Library Mean |
|---|---|---|---|
| SAT1 | 3.62 | 3.84 | 3.54 |
| SAT2 | 3.30 | 3.48 | 4.27 |
| SAT3 | 3.40 | 3.56 | 3.77 |
| SAT4 | 3.35 | 3.58 | 3.63 |
| SAT5 | 3.62 | 3.75 | 3.95 |
| SAT6 | 2.82 | 3.42 | 3.21 |
| SAT | 3.22 | 3.62 | 3.69 |

SAT: overall spatial satisfaction; SAT1: physical environment elements; SAT2: quality of interior design; SAT3: service facility availability; SAT4: furniture comfort and variety; SAT5: spatial diversity; SAT6: learning space controllability.

Users at WHU had the second-highest scores, with high scores for good space and physical environment, diversified learning space, and sufficient supply of spaces. Quality of interior design received a low score probably because the library space, facilities, and décor at WHU, which were rebuilt and refurbished a decade ago, are worn out.

The overall satisfaction score of users at the HUST library space was the lowest. The reconstruction of the HUST library included renovation upgrades and the addition of some reading and leisure study areas. However, no private study rooms, which students need, were included in the renovation. Overall, learning space controllability, quality of interior design, and furniture comfort and variety had the lowest scores at HUST.

### 4.3. Correlation Analysis and Linear Regression Analysis

Correlation analysis was conducted using SPSS software to analyze the data concerning overall satisfaction with the library spaces and six subitems concerning satisfaction with the library spaces. The results showed a correlation between all variables (as illustrated in Table 5).

**Table 5.** Correlation analysis of university library space satisfaction questionnaire.

|  | SAT | SAT1 | SAT2 | SAT3 | SAT4 | SAT5 | SAT6 |
|---|---|---|---|---|---|---|---|
| SAT | 1 |  |  |  |  |  |  |
| SAT1 | 0.459 ** | 1 |  |  |  |  |  |
| SAT2 | 0.238 ** | 0.335 ** | 1 |  |  |  |  |
| SAT3 | 0.581 ** | 0.441 ** | 0.609 ** | 1 |  |  |  |
| SAT4 | 0.496 ** | 0.385 ** | 0.591 ** | 0.772 ** | 1 |  |  |
| SAT5 | 0.529 ** | 0.427 ** | 0.671 ** | 0.668 ** | 0.622 ** | 1 |  |
| SAT6 | 0.528 ** | 0.391 ** | 0.504 ** | 0.616 ** | 0.608 ** | 0.658 ** | 1 |

** indicates that at the level of 0.01, the correlation is significant. SAT: overall spatial satisfaction; SAT1: physical environment elements; SAT2: quality of interior design; SAT3: service facility availability; SAT4: furniture comfort and variety; SAT5: spatial diversity; SAT6: learning space controllability.

To determine the importance of the influence of different factors on students' overall spatial satisfaction, six subitems of satisfaction were used as independent variables and overall spatial satisfaction was used as the dependent variable. An additional multiple linear regression analysis was conducted on the satisfaction data above using SPSS software. The independent sample *t*-test (independent sample *t*-test, a statistical technique that is used to analyze the mean comparison of two independent groups) on gender found that gender was not a variable that significantly influenced spatial satisfaction. A one-way ANOVA on education level, major discipline, and frequency of library visits found that all three had significant effects on spatial satisfaction. Education level, major discipline, and library visits were used as control variables.

The linear regression model fits well with $R^2$ (or R-squared, a statistical measure that represents the proportion of the variance for a dependent variable that is explained by an independent variable or variables in a regression model) = 0.51, and the five subitems of satisfaction can explain 51% of the variation of the overall satisfaction with the library space. This result indicates the influence of service facility availability, quality of interior design, physical environment elements, learning space controllability, and spatial diversity on the overall satisfaction of space reliably. The regression equation is significant, F (a ratio of two variances. Variances are a measure of dispersion or how far the data are scattered from the mean. Larger values represent greater dispersion) = 51.639, *p* (or *p*-value, a statistical measurement used to validate a hypothesis against observed data. A *p*-value measures the probability of obtaining the observed results, assuming that the null hypothesis is true. The lower the *p*-value, the greater the statistical significance of the observed difference) < 0.001. The supply of service facility availability significantly affects the overall satisfaction (beta = 0.551, $p < 0.05$); the quality of interior design significantly affects the overall satisfaction (beta = 0.384, $p < 0.05$); the physical environment elements significantly affect the overall satisfaction (beta = 0.384, $p < 0.05$) and significantly affects overall satisfaction (beta = 0.307, $p < 0.05$); self-control of learning space significantly affects overall satisfaction (beta = 0.228, $p < 0.05$); and spatial diversity significantly affects overall satisfaction (beta = 0.306, $p < 0.05$) (see Table 6).

**Table 6.** Correlation analysis of university library space satisfaction questionnaire.

| Model | Nonstandardized Coefficient | | Standardization Coefficient | t | Significance | Collinearity Statistics | |
|---|---|---|---|---|---|---|---|
| | B | Standard Error | Beta * | | | Tolerance | View |
| (Constant) | 0.201 | 0.228 | | −0.880 | 0.379 | | |
| SAT1 | 0.307 | 0.056 | 0.208 | 5.497 | 0.002 | 0.762 | 1.312 |
| SAT2 | 0.384 | 0.051 | −0.359 | −7.489 | 0.012 | 0.478 | 2.093 |
| SAT3 | 0.551 | 0.091 | 0.354 | 6.082 | 0.002 | 0.324 | 3.086 |
| SAT4 | 0.073 | 0.076 | 0.053 | 0.958 | 0.339 | 0.355 | 2.816 |
| SAT5 | 0.306 | 0.060 | 0.278 | 5.074 | 0.035 | 0.365 | 2.739 |
| SAT6 | 0.228 | 0.061 | 0.177 | 3.708 | 0.026 | 0.483 | 2.069 |

* (Beta: a "unit-free" measure of effect size, one that can be used to compare the magnitude of effects of predictors measured in different units) SAT: overall spatial satisfaction; SAT1: physical environment elements; SAT2: quality of interior design; SAT3: service facility availability; SAT4: furniture comfort and variety; SAT5: spatial diversity; SAT6: learning space controllability.

The regression equation is:

Overall satisfaction of space = 0.201 + 0.551 × service facility availability + 0.384 × quality of interior design + 0.307 × physical environment elements + 0.306 × spatial diversity + 0.228 × learning space controllability

## 5. Discussion

In this study, we used recently renovated libraries at three universities in Wuhan as research objects and 457 users as research subjects. We distributed questionnaires to

the subjects, which asked them to evaluate their university's library spaces and their satisfaction with the spaces. We then used statistical analysis methods to quantitatively analyze library users' answers concerning their evaluation of and satisfaction with their library's spaces. The results identified five major design factors that affect user satisfaction. Service facility availability was found to be the most important design factor. Other design factors included quality of interior design, physical environment elements, spatial diversity, and learning space controllability.

### 5.1. Service Facility Availability Is the Most Important Factor Affecting Students' Satisfaction with Space Renovation

The construction of information sharing space is indispensable in today's information technology era. Ubiquitous and easily accessible computer devices and network support are essential factors for enhancing student satisfaction. The learning profiles of contemporary college students—highly dependent on the Internet, experience oriented, and preference for multithreaded work and interaction—require college libraries to transform and respond rapidly. Long open hours are essential for library users who often need a period of immersive learning. In addition, the present study confirmed the importance of the availability of food as this amenity is a supportive service for students during long periods of study. Studies by Deng (2017) and Velusamy (2021) demonstrated that students prefer to study and interact with friends in places where food and water are available due to the strong connection between food and socialization. This is particularly the case for students who intend to study for long periods in the library and need timely energy replenishment while they study [16,20]. Therefore, it is recommended that the library extend its opening hours, provide light and fast-food services for students, increase the function of some café spaces, and improve the information network infrastructure.

### 5.2. Sophisticated Quality of Interior Design Can Effectively Enhance Students' Satisfaction with the Space

The generation born in the early 21st century has started higher education. They are accustomed to living in well-furnished environments from an early age, so a well-decorated interior is essential for the younger generation. This confirms the findings of a study by Jamieson (2003) that interior aesthetic elements, such as color scheme, flooring materials, and the quality and type of wall materials, have a considerable impact on individuals [30]. Oliveira et al. (2016) found that when the overall environment of a learning space is not attractive (i.e., when the furniture is worn, the décor bland and old-fashioned), it can significantly reduce the time students spend studying in the space [27]. A stylishly decorated space with a range of furniture, textures, and colors offers a rich, vivid, enjoyable, and refreshing experience that can stimulate curiosity, creativity, motivation, and intellectual ability.

### 5.3. Excellent Physical Environment Elements Enhance Students' Satisfaction with the Space

The findings of this study reaffirm that enhancing the physical attributes of the learning environment can improve student satisfaction with the space. Studies (Andrew, 2018; Lam 2019) indicate that adequate natural light and visibility within a space can enhance the attractiveness of the learning space, bring aesthetic pleasure to learners, and influence students' moods and attitudes [13,31]. Woolner (2007) ranked air temperature and air quality as the most important factors affecting student performance [22]. Empirical evidence from Wu (2021) has shown that temperature is the primary criterion for students' choice of informal learning spaces. Students prefer learning spaces with comfortable temperatures and also want to be able to adjust the temperature to their needs and for different times of the day [12].

### 5.4. Adequate Spatial Diversity Enhances Student Satisfaction

The primary type of library space has traditionally been the general-purpose reading room. Existing research findings indicate that students' two key learning activities in

libraries are individual learning activities and cooperative learning activities. Individual learning activities involve a highly focused process of knowledge internalization that requires a quiet environment—a type of learning space that is highly demanded by students. Learning spaces that are not soundproofed properly can negatively affect students' learning activities as excessive noise can lead to distraction and agitation [8,22]. Several reliable studies have also suggested that long-term noise exposure can impair cognitive function [22]. Most reading rooms do not guarantee a quiet environment; thus readers have to endure mutual disturbance and a lack of privacy. Therefore, libraries should increase the proportion of individual learning spaces by creating varied individual learning spaces ranging from private to semiopen to open.

Cooperative learning activities involve knowledge exchange and sharing even though the sounds of conversation disturb other learners. To facilitate cooperative learning that does not disturb individual learners, the library should increase the number of closed-group discussion rooms. Libraries can also expand the proportion of leisure learning areas by increasing the number and types of spaces where open communication and cooperative learning can occur—ranging from closed to semiopen to open spaces.

*5.5. Enhancing Learning Space Controllability Can Increase Students' Satisfaction with the Space*

The results of this study confirm that enhancing students' spatial control over their study area can improve user satisfaction. This includes students' ability to control and re-organize the following elements in their study space: external noise, layout, privacy, and intensity of the lighting. As students have different personalities, study habits, and learning styles, libraries need to enhance students' control over their study areas to meet their diverse needs. In future space renovations, the number of spaces that students can control on their own should be increased, and both private and public spaces should be available to users.

*5.6. Tapping into the Positive Role of Comfortable and Varied Furniture in Learning Space*

Although the design factor furniture variety did not have a significant effect on spatial satisfaction, it is necessary to conduct an in-depth follow-up study on this design factor. Due to the relatively short history of space renovation in China and the associated budgetary cost constraints, a lack of investment and emphasis on furniture persists. Furniture is not just inanimate blocks; it can be a medium for active participation in the collaborative learning process. Thus the integrated design of furniture and space should be a future research trend.

## 6. Conclusions

The theory of library space renovation was introduced to China in the early 2000s. More than a decade of theoretical and practical development followed, during which significant progress was achieved in the renovation of university library spaces. Through empirical research, we have determined, with suitable evidence, the key spatial elements that can effectively enhance user satisfaction. The conclusions of this study will provide more targeted design strategies for future library space:

(1)     *Service facility availability*

Our empirical research revealed that several library users desired storage space. Future designs can consider transforming walls into storage spaces or combining them with interior furniture to create additional storage space. In addition, furniture can be re-engineered to include more power outlets. The Wi-Fi signal quality can be improved using Wi-Fi signal boosters in areas with high usage and utilization rate based on the observations of the library managers. Electronic devices, such as printers, photocopiers, and other devices, can be set up in multiple brightly colored spaces with clear signs indicating their location. Furthermore, leisure spaces, such as cafes and light refreshment areas, can be added to the library. Finally, 24-h access to certain sections of the library can be considered for the benefit of students.

(2)   *Quality of interior design*

We, as a society, tend to decorate and beautify our surroundings and have developed several methods to improve interior design [32]. Decoration, like biophilia, plays a key role in generating comfort and well-being in the built environment. Decoration employs the smallest articulated scales to generate organized complexity. We "feed visually" on this organized complexity, which makes it a necessary component of our environment [33]. Thus, designers can use memorabilia related to the history of the university to decorate the space and enrich the design with details, similar to the interior design seen at the Firestone Library of Princeton University. Moreover, designers can use different decorative materials for different spaces depending on their function. For instance, the interior can be designed extensively with natural elements, such as wood, to create a relaxed learning atmosphere. In addition, the interior walls should have distinctive warm colors, such as yellow and red, which can increase the vitality of the space and inspire positive emotions in the inhabitants of the library. Shape wall surfaces engage us on a visceral level so that we feel at home in our environment [33].

(3)   *Physical environment elements*

In terms of internal acoustics, noise interference can be reduced by laying sound-absorbing materials, such as carpets on the floor, arranging greenery in the room, and setting up partitions. In the case of lighting, natural lighting should be provided by using wall windows and skylights as much as possible. Natural light is not merely essential to perceive and then to evaluate our surroundings; our skin requires sunlight in order to manufacture vitamin D, crucial to our metabolism [34]. In addition, as demonstrated in Hopkinson and Longmore's experiment, the local illumination of workbenches results in better concentration levels when compared with uniform background illumination [32]. Therefore, concentrated warm point light sources should be installed at study tables in a uniform artificial lighting environment to improve levels of concentration. In terms of ventilation quality, the number of window openings and window opening areas should strictly adhere to the corresponding standards to ensure sufficient indoor lighting and ventilation, priority should be given to green and environmentally friendly decorations and furniture, and green plants should be appropriately arranged in the library to improve the quality of the interior space. In terms of indoor temperature conditions, library managers should monitor the indoor temperature in real time and adjust it accordingly. In addition, the indoor climate environment should be properly regulated depending on the natural climate, instead of solely relying on equipment, such as air conditioners.

(4)   *Spatial diversity*

Learning spaces in libraries can be broadly divided into three categories: individual, group, and recreational learning spaces. For individuals and groups, the library should provide various kinds of spaces with different degrees of openness (i.e., completely private, semiopen, or completely open). Furthermore, the privacy of individual learning spaces should be improved using wall and glass enclosures, while the group learning spaces should have more interactive designs based on the combination of furniture and other equipment, such as whiteboards. Recreational learning spaces should be open with brightly colored furniture and decorations to enliven the space and create a relaxed and free learning environment (see Figure 3).

(5)   *Learning space controllability*

The design of library spaces should be student centered. However, learning styles, habits, and preferences differ among students. Thus, designers should ensure that the spaces can be modified by the users according to their preferences, thereby increasing their spatial autonomy and enhancing their sense of spatial territory and spatial belonging. For instance, in terms of lighting, users should be able to control the intensity, direction, and height of the lights, in addition to switching them on and off. In terms of acoustics and privacy, the library learning spaces should be divided into quiet areas and relatively livelier

ones. In addition, open learning spaces should use elements such as partitions, whiteboards, glasses, and furniture enclosures to enhance the sense of privacy while reducing noise interference between users. As regards temperature conditions, zone management should be applied to different learning spaces in the library. In other words, room temperatures should be suitably regulated to ensure that densely populated areas are maintained at lower room temperature than sparsely populated areas.

(6)   *Furniture comfort and variety*

Mehaffy (2015) argues that ample seating should be provided within the public realm, and at least some of the seating is movable so that people can adjust their position for comfort [33]. Owing to differences in size and height, people adopt different sitting postures. Therefore, learning spaces should not be furnished with identical seating [32]. In terms of comfort, furniture made of different soft materials should be added to ensure the physical comfort of students who are there to study for long periods. Furthermore, furniture should be diverse in terms of the privacy they offer; furniture with a high degree of enclosure can create a more private learning space, whereas furniture with a low degree of enclosure can be placed in open and recreational areas to create an atmosphere of open communication. Finally, furniture pieces that can be moved and freely assembled should be used as they can adapt to serve different functions. Consequently, students can use the furniture flexibly to suit their needs and different learning styles.

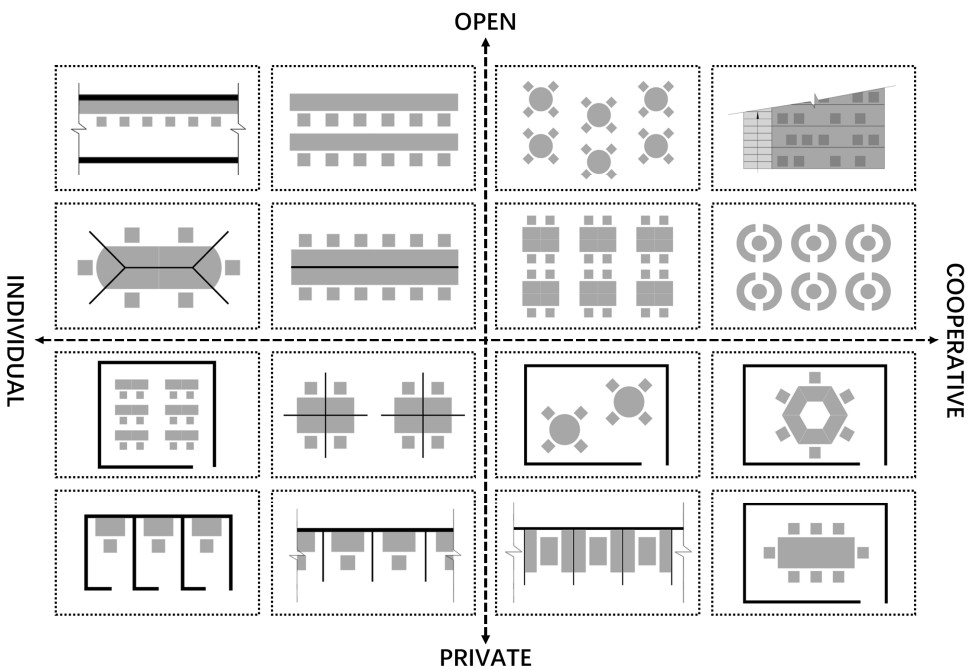

**Figure 3.** Design patterns of learning space.

**Author Contributions:** Conceptualization, L.P. and W.W.; methodology, W.W., L.P. and W.F.; software, W.W.; validation, L.P. and W.W.; formal analysis, L.P. and W.W.; investigation, W.W.; data curation, W.W., L.P. and W.F.; writing—original draft preparation, W.W., L.P. and W.F.; writing—review and editing, L.P., W.F., S.J. and Y.L.; visualization, W.F. and S.J.; supervision, L.P. and W.F.; project administration, L.P. All authors have read and agreed to the published version of the manuscript.

**Funding:** This research was supported by NSFC, grant number 51978294.

**Institutional Review Board Statement:** The study was conducted according to the guidelines of the Declaration of Helsinki, and approved by the Ethics Committee of the Tongji Medical College of Huazhong University of Science and Technology (protocol code S052, 27 April 2022).

**Informed Consent Statement:** Informed consent was obtained from all subjects involved in the study.

**Data Availability Statement:** Data is contained within this article.

**Conflicts of Interest:** The authors declare no conflict of interest.

**Appendix A. Questionnaire Survey on Library Space Satisfaction**

Hello, we are the research group of "Research on Spatial Value of University Library Based on User Experience." The purpose of this questionnaire survey is to comprehensively understand your needs and satisfaction of library learning space from the perspective of user experience so as to provide reference for optimizing the space design of the university library. We solemnly make an academic commitment to you and hope you can fill in the following with confidence:

This survey is conducted anonymously. The questionnaire information you filled in is only for academic research and will be kept strictly confidential.

1.   [single choice] Your gender is: ( )

☐   male
☐   female

2.   [single choice] You are: ( )

☐   undergraduate
☐   postgraduate student
☐   doctoral candidate
☐   faculty and researcher

3.   [single choice] Subject of your major: ( )

☐   law, philosophy, and economics
☐   pedagogy, literature, and history
☐   science, engineering, agriculture, and medicine
☐   military science, management science, and art science

4.   [single choice] On average, how often do you go to the library this semester: ( )

☐   every day
☐   weekly
☐   monthly
☐   quarterly
☐   basically not
☐   others

Please specify:

5.   [single choice] On average, the time you spend in the library each time is about: ( )

☐   less than 30 min
☐   30 min–1 h
☐   1–3 h
☐   more than 3 h
☐   others

Please specify:

6.   [multiple choice] Your main purpose of going to the library is to/for: ( )

☐   Self-study
☐   complete group work
☐   borrow and return books
☐   speech practice
☐   use electronic resources in the library for learning
☐   conduct social activities such as communication and interaction with others
☐   entertainment (watching movies, listening to music, playing games, etc.)
☐   others

Please specify:

7.      Your satisfaction with the following elements of the library on the main campus of Huazhong University of Science and Technology is:

| Evaluation Dimension | Evaluation Factor | Your Comments | | | | |
|---|---|---|---|---|---|---|
| | | **1** | **2** | **3** | **4** | **5** |
| | | **Very Dissatisfied** | **Somewhat Dissatisfied** | **Neither Satisfied nor Dissatisfied** | **Somewhat Satisfied** | **Very Satisfied** |
| Physical environment | Sufficient natural lighting | | | | | |
| | Good artificial lighting | | | | | |
| | Good ventilation | | | | | |
| | Suitable indoor temperature | | | | | |
| | Low noise interference | | | | | |
| Interior decoration | The overall decoration is highly exquisite | | | | | |
| | Good color matching for the overall decoration | | | | | |
| Service facility availability | Sufficient storage space | | | | | |
| | Wi-Fi signal quality is good | | | | | |
| | Sufficient supply of power sockets | | | | | |
| | Printing, copying, and other service are easy to use | | | | | |
| | The computer workstation is easy to use | | | | | |
| | Clear guiding signs | | | | | |
| | The supply of food and drinking water meets my demand | | | | | |
| | The opening time of the library is appropriate | | | | | |
| Furniture comfort and variety | Comfort of furniture | | | | | |
| | Variety of furnishings | | | | | |
| | Flexibility of furniture | | | | | |
| | Flexibility of indoor space layout | | | | | |
| Spatial flexibility | The supply of accessible study rooms and reading rooms is sufficient | | | | | |
| | There is an adequate supply of exclusive rooms for group discussion | | | | | |
| | The supply of exclusive personal learning rooms is sufficient | | | | | |
| | The supply of accessible leisure space is sufficient | | | | | |
| Learning space controllability | I can control the lighting level in my study area | | | | | |
| | I can control the noise level in my learning area | | | | | |
| | I can control the temperature of my study area | | | | | |
| | I can control the privacy of my study area | | | | | |

8.      Your overall satisfaction with the space of the library on the main campus of Huazhong University of Science and Technology is:

☐    very dissatisfied
☐    somewhat dissatisfied
☐    neither satisfied nor dissatisfied
☐    somewhat satisfied
☐    very satisfied

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
