# Peer review of "Student Experience and Satisfaction in Academic Libraries: A Comparative Study among Three Universities in Wuhan"

_buildings, doi:10.3390/buildings12050682_

Round 1
Reviewer 1 Report
In the result section, the scores between three University libraries were compared. As the authors write: “Users at WUT had the highest overall space satisfaction scores indicating that this recently constructed library met students’ learning needs.”. But as table 2 presents in WUT SAT 1 score (physical environment elements) was the lowest, compared to the two other universities with an overall score lower. A short comment/discussion on this will be recommended – why the highest scored library (WUT) was the lowest in the evaluation of physical environment elements, which are an important factor in users' satisfaction according to the study. Such reflections would be helpful for architects working on library space concepts.
My major concern is that some of the results are impossible or difficult to interpret because the authors have not defined the following symbols used in the statistical analysis: KMO, p, t-test, R2, F, beta.
They should be clearly defined or referenced. The regression model should also be presented. SPSS software should be referenced.
Figure 2. Floor plans of the libraries of the three universities – library halls/spaces are not marked on the 2nd floor.
Author Response
Point 1: In the result section, the scores between three University libraries were compared. As the authors write: “Users at WUT had the highest overall space satisfaction scores indicating that this recently constructed library met students’ learning needs.”. But as table 2 presents in WUT SAT 1 score (physical environment elements) was the lowest, compared to the two other universities with an overall score lower. A short comment/discussion on this will be recommended – why the highest scored library (WUT) was the lowest in the evaluation of physical environment elements, which are an important factor in users' satisfaction according to the study. Such reflections would be helpful for architects working on library space concepts.
Response 1: Thank you for your comments, and we have responded in lines 282 to 286 of the paper. Overall spatial satisfaction is measured based on multiple dimensions. The regression model at the end of the article shows that the effect of SAT1(physical environment elements) on overall spatial satisfaction in this study is less than that of SAT2(quality of interior design) and SAT3(service facility availability). WUT has the highest scores for both dimensions among the three universities, and therefore has a higher score for overall spatial satisfaction.
Point 2: My major concern is that some of the results are impossible or difficult to interpret because the authors have not defined the following symbols used in the statistical analysis: KMO, p, t-test, R2, F, beta. They should be clearly defined or referenced. The regression model should also be presented. SPSS software should be referenced.
Response 2: Thanks to your comments. We have added the definitions of the symbols in the footnotes on page 7 and 10 of the paper. And the regression model has been added to lines 338 to 341 of the paper.
Point 3: Figure 2. Floor plans of the libraries of the three universities – library halls/spaces are not marked on the 2nd floor.
Response 3: Thanks to your comments. We have added the modified floor plans to Table 1. Current status of learning spaces in three universities (line 221 of the paper).

Reviewer 2 Report
Libraries have been centers of knowledge since the beginning of human culture. The term university library contains an obvious relationship between knowledge preservation and education. University libraries functioned as a space for study-learning from the very beginning of their existence, only the form of using the library as an environment for study changed. The authors of the article analyze a comparison of the current state of three university libraries in Wuhan. If the authors work with the concept of space, specifically inner space, at least references to the theory and philosophy of space would be required, such as elementary studies by Gaston Bachelard and Bruno Zevi. A bit of space philosophy fits the university library. According to the authors, the analyzes sound too technocratic. However, nor technical education cannot be narrowed in this way either. Microclimatic parameters are physically measurable. The questionnaire method is also suitable for determining the immeasurable aesthetic properties of space. The authors improperly use the term Interior Decoration repeatedly in the text, for example in the title of subchapter 5.2., Line 361. From the point of view of professional terminology, it would be more appropriate to use quality of interior design. User-quality space needs clear aesthetic parameters. The authors could process an abstract spatial diagram of the three libraries showing the operation. Thus, it would be possible to compare the quality of spatial concepts and relate them to students' answers. For inspiration e.g. fig. 11. in: https://www.researchgate.net/publication/312263676_Evolutionary_approach_for_spatial_architecture_layout_design_enhanced_by_an_agent-based_topology_finding_system The diagram could be located at line 90 or 217. Finally, there is a lack of a summary of research potential and, in particular, I miss a generalization for future library designs.Author Response
Point 1: Libraries have been centers of knowledge since the beginning of human culture. The term university library contains an obvious relationship between knowledge preservation and education. University libraries functioned as a space for study-learning from the very beginning of their existence, only the form of using the library as an environment for study changed. The authors of the article analyze a comparison of the current state of three university libraries in Wuhan. If the authors work with the concept of space, specifically inner space, at least references to the theory and philosophy of space would be required, such as elementary studies by Gaston Bachelard and Bruno Zevi. A bit of space philosophy fits the university library.
Response 1: Thank you for your comments. The philosophy of space you mentioned is the theoretical basis of our research and it has helped us a lot. We have added the relevant theory to lines 51 to 57 of the paper.
Point 2: According to the authors, the analyzes sound too technocratic. However, nor technical education cannot be narrowed in this way either. Microclimatic parameters are physically measurable. The questionnaire method is also suitable for determining the immeasurable aesthetic properties of space.
Response 2: Thank you for your comments. Our study only measured users' psychological satisfaction with the library space, not the physical space aspect. And based on your comments, we will optimize our research methodology in future studies
Point 3: The authors improperly use the term Interior Decoration repeatedly in the text, for example in the title of subchapter 5.2., Line 361. From the point of view of professional terminology, it would be more appropriate to use quality of interior design.
Response 3: Thank you for your comments. We have corrected the inappropriate expressions in line 279,285,288,294,323,327,339,351,370 of the paper.
Point 4: User-quality space needs clear aesthetic parameters. The authors could process an abstract spatial diagram of the three libraries showing the operation. Thus, it would be possible to compare the quality of spatial concepts and relate them to students' answers. For inspiration e.g. fig. 11. in: https://www.researchgate.net/publication/312263676_Evolutionary_approach_for_spatial_architecture_layout_design_enhanced_by_an_agent-based_topology_finding_system The diagram could be located at line 90 or 217.
Response 4: Thank you for your comments. We have added the abstract spatial diagram of three libraries to Table 1 in line 221 of the paper. In addition, we have added the design pattern diagram of the library spaces(Figure 3, line 514) in the conclusion of the paper.
Point 5: There is a lack of a summary of research potential and, in particular, I miss a generalization for future library designs.
Response 5: Thank you for your comments. Based on your comments, we have expanded the conclusions (line 432 to 515) of the paper to propose strategies for future library design.

Round 2
Reviewer 1 Report
The authors have answered my comments and revised the manuscript accordingly. I have no further comments.Author Response
Your comments have helped us a lot. Thank you for reviewing my manuscript!